# Heparin-induced tau filaments are polymorphic and differ from those in Alzheimer's and Pick's diseases

**Wenjuan Zhang, Benjamin Falcon, Alexey G Murzin, Juan Fan, R Anthony Crowther, Michel Goedert*, Sjors HW Scheres***

MRC Laboratory of Molecular Biology, Cambridge, United Kingdom

**Abstract** Assembly of microtubule-associated protein tau into filamentous inclusions underlies a range of neurodegenerative diseases. Tau filaments adopt different conformations in Alzheimer's and Pick's diseases. Here, we used cryo- and immuno- electron microscopy to characterise filaments that were assembled from recombinant full-length human tau with four (2N4R) or three (2N3R) microtubule-binding repeats in the presence of heparin. 2N4R tau assembles into multiple types of filaments, and the structures of three types reveal similar 'kinked hairpin' folds, in which the second and third repeats pack against each other. 2N3R tau filaments are structurally homogeneous, and adopt a dimeric core, where the third repeats of two tau molecules pack in a parallel manner. The heparin-induced tau filaments differ from those of Alzheimer's or Pick's disease, which have larger cores with different repeat compositions. Our results illustrate the structural versatility of amyloid filaments, and raise questions about the relevance of in vitro assembly.

DOI: https://doi.org/10.7554/eLife.43584.001

*For correspondence:
mg@mrc-lmb.cam.ac.uk (MG);
scheres@mrc-lmb.cam.ac.uk
(SHWS)

Competing interest: See
page 19

Reviewing editor: Nikolaus
Grigorieff, Janelia Research
Campus, Howard Hughes
Medical Institute, United States

## Introduction

The ordered assembly of tau protein into amyloid filaments defines a number of neurodegenerative diseases, also known as tauopathies; they are the most common proteinopathies of the human nervous system (*Goedert et al., 2017*). The physiological function of tau is to promote the assembly, and possibly stability, of microtubules. Free tau is intrinsically disordered, but the microtubule-binding repeats (R1-4) and some adjoining sequences adopt structure when bound to microtubules (*Al-Bassam et al., 2002*; *Kellogg et al., 2018*).

Six tau isoforms ranging from 352 to 441 amino acids are expressed in adult human brain from a single *MAPT* gene (*Goedert et al., 1989*). They differ by the presence or absence of inserts of 29 or 58 amino acids (encoded by exons 2 and 3, with exon three being only transcribed in conjunction with exon 2) in the N-terminal half, and the inclusion, or not, of the second 31 amino acid microtubule-binding repeat (R2), encoded by exon 10, in the C-terminal half. Inclusion of exon 10 results in the production of three isoforms with four repeats (4R) and its exclusion in a further three isoforms with three repeats (3R). The repeats comprise residues 244–368 of tau, in the numbering of the 441 amino acid isoform (2N4R tau).

Tau filaments in some neurodegenerative diseases have different isoform compositions (*Goedert et al., 2017*). For example, a mixture of all six isoforms is present in the tau filaments of Alzheimer's disease (AD), but in progressive supranuclear palsy (PSP) tau filaments are made of 4R tau, whereas the filaments of Pick's disease (PiD) are made of 3R tau. The repeats make up the filament cores, with the remainder of tau forming the fuzzy coat (*Wischik et al., 1988*).

The determination of the cryo-EM structures of tau filaments from the brains of individuals with AD (*Falcon et al., 2018a*; *Fitzpatrick et al., 2017*) and PiD (*Falcon et al., 2018b*) revealed the

atomic structures of their cores. In AD, two types of filaments are observed: paired helical filaments (PHFs) and straight filaments (SFs) (*Crowther, 1991*). PHFs and SFs are ultrastructural polymorphs, because they consist of two identical protofilaments with a C-shaped core, but differ in the interfaces between protofilaments. The core is made of amino acids 273/304–380, comprising the carboxyl-terminal parts of R1 and R2, all of R3 and R4, as well as part of the C-terminal domain. This is consistent with the presence of all six tau isoforms in AD filaments (*Goedert et al., 1992*).

In PiD, narrow and wide filaments (NPFs and WPFs) are seen. NPFs are made of a single protofilament with an elongated J-shaped core comprising amino acids 254–378 of 3R tau. WPFs are made of two NPFs. The presence of the C-terminal region of R1 (residues 254–274) in the core of NPFs and WPFs explains why only 3R tau is present in PiD filaments. The markedly different structures of the protofilament cores from AD and PiD established the existence of distinct molecular conformers of aggregated tau in different human tauopathies.

Several methods have been used for the in vitro study of tau filaments. Initial work showed that the repeat region of tau formed filaments (*Crowther et al., 1992*; *Wille et al., 1992*). Full-length tau assembled upon addition of polyanionic molecules, such as RNA (*Kampers et al., 1996*), polyglutamate (*Friedhoff et al., 1998*) and fatty acids (*Wilson and Binder, 1997*) in micellar form (*Chirita et al., 2003*). Prior to the above, heparin and other sulphated glycosaminoglycans had been shown to induce bulk assembly of recombinant full-length tau into filaments (*Goedert et al., 1996*; *Pérez et al., 1996*). This led to a conformational change from a mostly random coil to a β-sheet structure in a region of the repeats containing hexapeptide motifs necessary for assembly (*Berriman et al., 2003*; *Hasegawa et al., 1997*; *von Bergen et al., 2000*; *von Bergen et al., 2001*). Heparin was subsequently used in many studies, including those aimed at understanding the effects of *MAPT* mutations on filament assembly (*Arrasate et al., 1999*; *Goedert et al., 1999*), and at identifying tau aggregation inhibitors (*Nacharaju et al., 1999*; *Pickhardt et al., 2005*; *Taniguchi et al., 2005*).

Although low-resolution, negative stain EM images have often been used to claim that heparin-induced tau filaments resemble those from human brain, an increasing number of studies has suggested that structural differences may exist between them, as outlined below. Tau filaments assembled from human recombinant 0N4R P301S tau and heparin had a reduced seeding activity when compared with that of sarkosyl-insoluble tau from the brains of mice transgenic for human 0N4R P301S tau (*Falcon et al., 2015*). Additional experiments showed that the conformational properties of the seed determined the properties of the seeded aggregates. Negative-stain EM, circular dichroism and chemical denaturation revealed differences between tau filaments that were seeded from AD-derived brain extracts and those that were assembled from recombinant tau using heparin (*Morozova et al., 2013*). Moreover, R2 and R3 are ordered in heparin-induced filaments of 4R tau (*Li et al., 2002*; *Mukrasch et al., 2005*; *Sibille et al., 2006*), whereas only R3 is ordered in 3R tau filaments (*Andronesi et al., 2008*; *Daebel et al., 2012*). This is in agreement with site-directed spin labelling combined with electron paramagnetic resonance spectroscopy, which showed that 3R and 4R heparin-induced tau filaments are different, but share a highly ordered structure in R3 (*Siddiqua and Margittai, 2010*). Double electron-electron resonance (DEER) spectroscopy has suggested that the structures of heparin-induced recombinant tau filaments are different from those of AD (*Fichou et al., 2018*).

Here, we used cryo-EM to determine the structures of heparin-induced filaments assembled from recombinant full-length 4R tau (2N4R, 441 amino acid isoform) and the corresponding 3R isoform (2N3R, 410 amino acid isoform) to resolutions sufficient for de novo atomic modelling. Immuno-EM confirmed which microtubule-binding repeats were in the ordered cores. Our results show that heparin-induced 2N4R filaments consist of a mixture of at least four different conformations, whereas heparin-induced 2N3R filaments mainly adopt a single conformation. All four resolved structures are different from those of the tau filaments in AD and PiD. These findings indicate that tau filament structures are even more versatile than previously thought, and illustrate how EM can be used to compare the structures of tau filaments from model systems with those formed in disease.

## Results

### Comparative morphology of heparin-induced 2N4R and 2N3R tau filaments

For 2N4R tau, we distinguished at least four different types of filaments in raw cryo-EM micrographs and in 2D class averages, (*Figure 1A*). We named the two most common types *snake* (~45%) and *twister* (~30%). Snake filaments have a crossover distance of 650 Å and vary in width between 40 and 100 Å. They display a sigmoidal curvature pattern, with deviations of up to 70 Å from a hypothetical straight line through their centre. Twister filaments have an almost constant width of 80 Å, and a crossover distance that is approximately two times shorter (~250 Å) than that of the snake filaments. We called the least common types of filaments *hose* (~20%) and *jagged* (~5%). Hose filaments display a sigmoidal curvature pattern that is less regular and has much wider curves compared to snake filaments, and hose filaments often appear to stop twisting. Jagged filaments are straighter than snake filaments, and are named after their somewhat rugged appearance around the edges, with filament widths ranging from 50 to 90 Å. They have a crossover distance of approximately 450 Å. In some micrographs, we observed filaments that change from one type into another (*Figure 1—figure supplement 1*). We observed the following transitions: twister to snake; twister to jagged; hose to snake and hose to jagged. Similar changes were also observed for tau filaments from AD brains and PiD brains, where a continuous filament could transition from a PHF to a SF, and from a NPF to a WPF, respectively (*Crowther, 1991*; *Falcon et al., 2018b*).

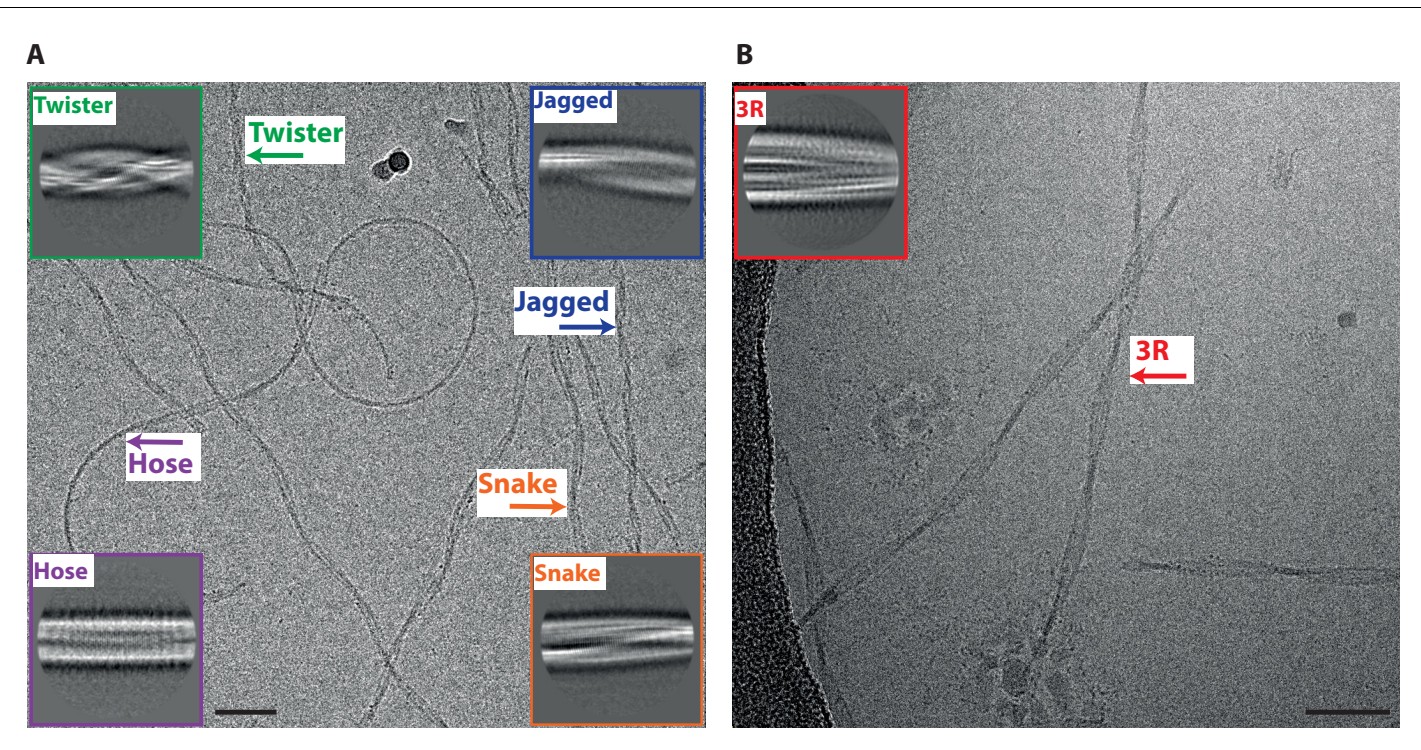

**Figure 1.** Different types of heparin-induced tau filaments. (**A**) Cryo-EM image of heparin-induced 2N4R tau filaments. (**B**) Cryo-EM image of heparin-induced 2N3R tau filaments. 2D class averages of each filament type are shown as insets. Scale bars, 50 nm.

DOI: https://doi.org/10.7554/eLife.43584.002

The following figure supplements are available for figure 1:

**Figure supplement 1.** Heparin-induced tau filaments can change from one type into another.
DOI: https://doi.org/10.7554/eLife.43584.003
**Figure supplement 2.** Cryo-EM image of heparin-induced filaments assembled from a mixture of 2N4R and 2N3R tau.
DOI: https://doi.org/10.7554/eLife.43584.004

Reference-free 2D class averaging of manually selected data sets for the four types confirmed the differences between them (*Figure 1A*-inset). For snake, twister and jagged filaments, 3D reconstructions led to maps with separated β-strands along the helical axis, which allowed for de novo atomic modelling (see below). However, 3D reconstruction failed for hose filaments, possibly because of a large degree of bending and an apparent lack of twisting in many segments.

Heparin-induced 2N3R tau filaments were more homogeneous than their 2N4R counterparts (*Figure 1B*): almost all filaments displayed a minor sigmoidal curvature pattern, with a crossover distance of 800 Å, and widths varying from 50 to 120 Å. Only 2% of filaments appeared to be wider (up to 160 Å). Reference-free 2D class averaging of the resulting data set confirmed the original observation that 2N3R tau filaments were different from the four 2N4R tau filament types (*Figure 1B*-inset). 3D reconstruction of the narrow 2N3R filaments led to a map with separated β-strands along the helical axis and sufficient resolution for de novo atomic modelling (see below).

Since tau filaments of AD consist of a mixture of 3R and 4R isoforms, we also performed heparin-induced in vitro assembly with equimolar amounts of 2N4R and 2N3R tau. A mixture of the same five filament types described above was observed (*Figure 1—figure supplement 2*). Although no co-assemblies with different morphologies were present, we cannot exclude the possibility that filaments with a mixed composition can adopt the same morphologies as observed for the homogeneous samples.

## Cryo-EM structure of 2N4R tau snake filaments

Of all heparin-induced tau filaments, the best helical reconstruction was obtained for the snake. The structure has a helical twist of −1.26°, an overall resolution of 3.3 Å, and a clear separation of β-strands along the helical axis (*Figure 2*; *Figure 2—figure supplement 1*). A pronounced offset of the centre of the ordered core from the helical axis explains the sigmoidal appearance of the filaments in projection (*Figure 2B*). The map allowed unambiguous de novo atomic modelling (*Figure 2*, Methods). The ordered core of the snake filament comprises residues 272–330, i.e. the last three residues of R1, all of R2 and most of R3 (*Figure 2A*). In the core, there are six β-strands, three from R2: β1 (274–280), β2 (282–291) and β3 (295–298), and three from R3: β4 (305–310), β5 (313–321) and β6 (327–330). Four of these strands form two stacks of cross-β, with β1 packing against β5 and β2 against β4. Both interfaces have mixed compositions of polar and hydrophobic groups (*Figure 2C, D*).

Both cross-β stacks are connected at an angle by short arcs, shaping the overall structure into a kinked hairpin (*Figure 2E,F*). The angle between β4 and β5 is 69° (Figure 7C), as measured from the coordinates of the Cα atoms of their first and last residues in the plane perpendicular to the helical axis. The inner corner of the kink is formed by K281 and L282, with both residues pointing towards the inside of the core; the outer corner is formed by Y310 and K311, both of which point outwards. A broad, hammerhead-like arc, harbouring β3, connects β2 to β4. Weaker density, which is still sufficiently well-defined for unambiguous tracing of the main chain, suggests that this arc is more flexible than the rest of the core. The [290]KCGSKD[295] motif, which adopts a very similar conformation to the homologous [353]KIGSLD[358] and [259]KIGSTE[263] motifs in the tau filament structures from AD and PiD, respectively (*Falcon et al., 2018b*; *Fitzpatrick et al., 2017*), connects β2 and β3. On the other side of the hammerhead arc, the [301]PGGG[304] motif forms part of a turn that connects β3 and β4. The hammerhead arc represents a new type of chain direction reversal in tau filaments, different from the triangular β-helix of the Alzheimer fold (*Fitzpatrick et al., 2017*), and the tighter U-turns of the Pick fold (*Falcon et al., 2018b*). The other end of the kinked hairpin is capped by the [322]CGSLG[326] motif, which brings C-terminal glycine 326 into close contact with N-terminal glycines 272 and 273. It is followed by a short C-terminal β-strand (β6), which is exposed to solvent on both sides, reminiscent of the C-terminus of the Pick fold (*Falcon et al., 2018b*).

In the core of snake filaments, there are 12 positively charged amino acids (9 lysines and three histidines), but only three negatively charged residues (all aspartates). The aspartates appear to form salt bridges: D283 to K280 on the inside corner of the kink; D295 to K290 inside the hammerhead arc; and D314 to K281 in the cross-β interfaces of β5 and β1. Most of the positively charged sidechains are exposed on the filament surface, where they face diffuse external densities (*Figure 2B*, indicated by yellow arrows), presumably corresponding to negatively charged groups of heparin. In addition, there is a significant external density covering a large exposed hydrophobic patch on the filament surface, formed by V306, I308 and Y310 (*Figure 2B*, indicated by a pink arrow).



**Figure 2.** Cryo-EM structure of 2N4R tau snake filaments. (**A**) β-strands and loop regions in the filaments are shown in different colours below the primary sequence of the microtubule-binding repeats (R1–R4). (**B**) Central slice of the 3D map. The position of the helical axis is indicated by a red cross, extra densities close to outward-facing lysines by yellow arrows, and extra density in front of the hydrophobic patches by a pink arrow. (**C**) Cryo-EM density with the atomic model. The sharpened, high-resolution map is in blue, and an unsharpened, 4.0 Å low-pass filtered map in grey. (**D**) Schematic view of the snake filament. (**E**) Rendered view of secondary structure elements in three successive rungs. (**F**) As in E, but in a view perpendicular to the helical axis.

DOI: https://doi.org/10.7554/eLife.43584.005

The following figure supplement is available for figure 2:

**Figure supplement 1.** Fourier shell correlation curves and side views of the 3D reconstruction of 2N4R tau snake filaments.

DOI: https://doi.org/10.7554/eLife.43584.006

## Cryo-EM structure of 2N4R tau twister filaments

The twister filament reconstruction, with an overall resolution of 3.3 Å and a helical twist of −3.38°, also allowed de novo atomic modelling (*Figure 3*; *Figure 3—figure supplement 1*). The ordered core of the twister filament is smaller than that of the snake filament, and is positioned much closer to the helical axis (*Figure 3B*). The ordered core comprises only residues 274–321 (*Figure 3A*), i.e. the last residue of R1, all of R2 and half of R3. As in the snake filament, these are organised into a kinked hairpin structure of two cross-β stacks connected by a hammerhead arc (*Figure 3C,D*). In the core, there are four β-strands, two from R2: β1 (274–284) and β2 (286–291), and two from R3: β3 (305–310) and β4 (313–321). The twister shares secondary structure with the snake for the ordered part of R3, with Y310 and K311 forming the outer corner of the kink, which is more bent (with an angle between β3 and β4 of 72°) (Figure 7C) than in the snake. In contrast, on the R2 side of the hairpin, the inner corner of the kink is formed by S285 and N286, extending β1 and shortening β2, compared to the snake filament. The change of R2 secondary structure in the twister filament reverses the interior/exterior orientations of four residues: the polar side-chains of D283 and S285 point

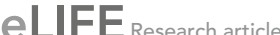

**Figure 3.** Cryo-EM structure of 2N4R tau twister filaments. (**A**) β-strands and loop regions in the filaments are shown in different colours below the primary sequence of the microtubule-binding repeats (R1–R4). (**B**) Central slice of the 3D map. The position of the helical axis is indicated by a red cross, extra densities close to the outward-facing lysines by yellow arrows, and extra density in front of the hydrophobic patches by pink arrows. (**C**) Cryo-EM density with the atomic model. The sharpened, high-resolution map is in blue, and an unsharpened, 4.0 Å low-pass filtered map in grey. (**D**) Schematic view of the twister filament. (**E**) Rendered view of secondary structure elements in three successive rungs. (**F**) As in E, but in a view perpendicular to the helical axis.

DOI: https://doi.org/10.7554/eLife.43584.007

The following figure supplement is available for figure 3:

**Figure supplement 1.** Fourier shell correlation curves and side views of the 3D reconstruction of 2N4R tau twister filaments.
DOI: https://doi.org/10.7554/eLife.43584.008

towards the inside of the core, whereas hydrophobic residues L282 and L284 point outwards. This results in the twister filament core having a more polar interior, and a more hydrophobic exterior than that of the snake. It also results in a repacking of both cross-β interfaces. Again, the density of the residues that form the hammerhead arc is weaker than that of the rest of the structure. The lower quality of the map in this region led to ambiguity in how the strands on opposite sides of the core connected to each other (*Figure 3E,F*). Therefore, we chose not to build an atomic model of residues 293–303 in this part of the structure.

## Cryo-EM structure of 2N4R tau jagged filaments

At 3.5 Å, the overall resolution for the jagged filament structure was lower than for the other two 4R filament types and the side-chain densities were resolved less well than for snake and twister filaments. Still, aided by the models for the other 4R tau filament structures, the map was of sufficient quality to propose an atomic model (*Figure 4*; *Figure 4—figure supplement 1*). The helical twist of −2.03° for the jagged structure falls between those of the snake and twister filaments (*Table 1*); the offset of the centre of the packing unit from the helical axis also lies in-between (*Figure 4B*). The extent of the ordered core, comprising residues 274–321, is almost identical to that of twister filament (*Figure 4A*). There are three β-strands, β1 (274–290), β2 (305–310) and β3 (313–321). Again, the structure is characterised by an overall kink, which, with an angle of 52° between β2 and β3 (Figure 7C), is the least pronounced of the three 4R tau filament structures (*Figure 4C,D*).

As in the other two types, the ordered part of R3 shares the same secondary structure, with Y310 and K311 forming the outer corner of the kink. However, unlike in the snake and twister filaments, there is no inner corner of the kink. Instead, one long, slightly bent β-strand extends from residues 274–290. Compared to the snake filament, the pair of adjacent lysine residues changes to interior/exterior orientations. The K280 side-chain points towards the inside of the core and makes a salt bridge with D314, whereas the K281 side-chain points outwards, and forms a salt bridge with D283. The strand residues on the N-terminal side of K280 have also their orientations reversed, and form a new cross-β interface with the β3 strand of R3. In contrast, the C-terminal half of this strand (residues 282–290) forms essentially the same cross-β interface with the β2 strand of R3, as in snake filaments. A hammerhead arc connecting these two strands may be similar to that in the snake. As in the other 4R tau filament structures, its density is weaker than that of the rest of the core, suggesting increased flexibility. The density of the whole arc was not good enough for model building, resulting in the omission of residues 291–303 from the atomic model, and ambiguity in how the strands on opposite sides of the core connected to each other (*Figure 4E,F*).

## Cryo-EM structure of 2N3R tau filaments

At an overall resolution of 3.7 Å, and with clear separation of β-strands along the helical axis, cryo-EM reconstruction of the 2N3R tau filaments allowed unambiguous de novo atomic modelling (*Figure 5*; *Figure 5—figure supplement 1*). With a helical twist of −1.05°, the 3R filament twists less than any of its 4R counterparts (*Table 1*). Again, an offset of the centre of the packing unit from the helical axis explains the sigmoidal patterns observed in projection (*Figure 5B*). The structure is strikingly different from those of the 2N4R tau filaments. Whereas all three 4R filaments contain a single molecule of tau per rung in the β-sheet, the 3R filament core contains two tau molecules on each rung. There exists no exact symmetry between the two molecules, which are arranged in a parallel cross-β packing (*Figure 5E,F*). In one molecule, residues 274–330 are ordered; in the other, residues 272–330 are ordered, i.e. the structured core comprises only the last residues of R1, and most of the residues of R3 (this being a 3R tau isoform, residues 275–305 of R2 are not present). At the N-terminal end of the two molecules, residues 274–310 form the first parallel β-strands (β1). The two tyrosines at the end of these strands point towards each other, which marks the beginning of a wider gap between the two molecules, which comprises in each K311, P312 and a small β-strand formed by residues 313–315 (β2). After that, the β-strands comprising residues 317–325 (β3) of both molecules come close together, to engage in tight and interdigitating cross-β packing, provided by the side-chains of S320, C322, S324 and the backbone of G326. At the C-terminal end of the ordered core of each tau molecule, residues 328–330 form a β-strand that faces away. It is equivalent to the C-terminal strand of the snake filaments, and likewise is exposed to solvent on both sides.

**Figure 4.** Cryo-EM structure of 2N4R tau jagged filaments. (**A**) β-strands and loop regions in the filaments are shown in different colours below the primary sequence of the microtubule-binding repeats (R1–R4). (**B**) Central slice of the 3D map. The position of the helical axis is indicated by a red cross, extra densities close to the outward-facing lysines by yellow arrows, and extra density in front of hydrophobic patches by a pink arrow. (**C**) Cryo-EM density with the atomic model. The sharpened, high-resolution map is in blue, and an unsharpened, 4.0 Å low-pass filtered map in grey. (**D**) Schematic view of the jagged filament. (**E**) Rendered view of the secondary structure elements in three successive rungs. (**F**) As in E, but in a view perpendicular to the helical axis.

DOI: https://doi.org/10.7554/eLife.43584.009

The following figure supplement is available for figure 4:

**Figure supplement 1.** Fourier shell correlation curves and side views of the 3D reconstruction of 2N4R tau jagged filaments.

DOI: https://doi.org/10.7554/eLife.43584.010

## Immuno-EM supports the atomic models

In previous studies, we used immuno-EM to confirm which microtubule-binding repeats form part of the ordered core of tau filaments from AD and PiD brains (*Falcon et al., 2018a*; *Falcon et al., 2018b*; *Fitzpatrick et al., 2017*). Epitopes of repeat-specific, anti-tau antibodies that are buried in the cores of tau filaments are not accessible, while epitopes located in the fuzzy coat are labelled. Moreover, pronase removes the fuzzy coat, which abolishes this positive labelling. We applied the same methods to corroborate the cryo-EM structures of heparin-induced filaments of 2N4R and 2N3R tau (*Figure 6*). We used antibodies specific for residues 1–16 at the N-terminus (BR133); 244–257 in R1 (BR136); 275–291 in R2 (Anti4R); 323–335 in R3 (BR135); 354–369 in R4 (TauC4); and 428–

**Table 1.** Cryo-EM structure determination and model statistics

| | 4 R-s | 4 R-t | 4 R-j | 3R |
|---|---|---|---|---|
| **Data collection and processing** | | | | |
| Microscope | Polara | Polara | Polara | Titan Krios |
| Voltage (kV) | 300 | 300 | 300 | 300 |
| Detector | Falcon-III | Falcon-III | Falcon-III | K2 (post-GIF) |
| Electron exposure (e–/Å$^2$) | 50 | 50 | 50 | 50 |
| Defocus range (µm) | −1.7 to −2.8 | −1.7 to −2.8 | −1.7 to −2.8 | −0.8 to −2.2 |
| Pixel size (Å) | 1.38 | 1.38 | 1.38 | 1.04 |
| Initial particle images (no.) | 303,754 | 187,555 | 44,456 | 788,359 |
| Final particle images (no.) | 52,441 | 141,461 | 35,695 | 149,909 |
| Map resolution (Å) | 3.3 | 3.3 | 3.5 | 3.7 |
| Helical rise (Å) | 4.70 | 4.70 | 4.70 | 4.70 |
| Helical twist (°) | −1.26 | −3.38 | −2.03 | −1.05 |
| Refinement | | | | |
| Map sharpening $B$ factor (Å$^2$) | −41.26 | −58.51 | −33.2 | −95.9 |
| Model composition<br>Non-hydrogen atoms<br>Protein residues | 1302 | 846 | 816 | 1218 |
| | 177 | 111 | 105 | 162 |
| R.m.s. deviations<br>Bond lengths (Å)<br>Bond angles (°) | 0.0094 | 0.0102 | 0.0099 | 0.0209 |
| | 0.9007 | 1.0727 | 1.1342 | 1.0457 |
| Validation<br>MolProbity score<br>Clashscore<br>Poor rotamers (%) | 1.56 | 1.92 | 1.13 | 1.65 |
| | 1.49 | 7.3 | 1.74 | 4.78 |
| | 1.96 | 0 | 0 | 0 |
| Ramachandran plot<br>Favored (%)<br>Allowed (%)<br>Disallowed (%) | 92.98 | 90.91 | 96.77 | 94.0 |
| | 100 | 100 | 100 | 98.0 |
| | 0 | 0 | 0 | 2 |
| EMPIAR | 10243 | 10243 | 10243 | 10242 |
| EMDB | 4563 | 4564 | 4565 | 4566 |
| PDB | 6QJH | 6QJM | 6QJP | 6QJQ |

DOI: https://doi.org/10.7554/eLife.43584.011

441 at the C-terminus (BR134). All antibodies labelled bands on Western blots of recombinant proteins (*Figure 6—figure supplement 1*).

As expected, heparin-induced 2N4R and 2N3R tau filaments were labelled by BR133 and BR134 before, but not after, pronase treatment. Similarly, BR136 and TauC4 decorated 2N4R and 2N3R filaments before, but not after, pronase treatment. By contrast, Anti4R and BR135 did not decorate 2N4R tau filaments, either before or after pronase treatment. BR135 also failed to decorate 2N3R tau filaments. This suggests that the N- and C-termini of tau, as well as the N-terminus of R1 and the C-terminus of R4 are in the fuzzy coat of all heparin-induced filaments, whereas R2 and R3 in 2N4R, and R3 in 2N3R tau filaments are in their ordered cores. These findings are consistent with the tau sequences observed in the core structures of 2N4R and 2N3R filaments.

**Figure 5.** Cryo-EM structure of 2N3R tau filaments. (**A**) β-strands and loop regions in the filaments are shown in different colours below the primary sequence of the microtubule-binding repeats (R1–R4). (**B**) Central slice of the 3D map. The position of the helical axis is indicated by a red cross, extra densities close to outward-facing lysines by yellow arrows, and extra density in front of hydrophobic patches by pink arrows. (**C**) Cryo-EM density with the atomic model. The sharpened, high-resolution map is in blue, an unsharpened, 4.0 Å low-pass filtered map in grey. (**D**) Schematic view of 2N3R tau filament. (**E**) Rendered view of the secondary structure elements in three successive rungs. (**F**) As in E, but in a view perpendicular to the helical axis.

DOI: https://doi.org/10.7554/eLife.43584.012

The following figure supplement is available for figure 5:

**Figure supplement 1.** Fourier shell correlation curves and side views of the 3D reconstruction of 2N3R tau filaments.

DOI: https://doi.org/10.7554/eLife.43584.013

## Discussion

Heparin-induced filaments of 2N4R tau are polymorphic, adopting at least four different conformations. Cryo-EM structures of three of these conformations reveal a common, kinked hairpin fold, with differences in kink, helical twist and offset distance of the ordered core from the helical axis. A 20° variation of the kink angle between the shared R3 β-strands 305–310 and 313–321 (from 52° to 72°) (*Figure 7*) may result from the optimisation of their cross-β packing interfaces with different R2 counterparts, and requires only minor adjustments of local backbone conformations of the $^{310}$YKP$^{312}$ turn. Both helical twist and offset distance from the helical axis may result from the optimisation of local interactions within and between the constituent β-sheets. For example, the larger helical twist



**Figure 6.** Immuno-EM of heparin-induced 2N4R and 2N3R tau filaments. (**A**) Schematic of 2N4R tau with N-terminal inserts (N1 and N2) and microtubule-binding repeats (R1, R2, R3, R4) highlighted. The epitopes of antibodies BR133 (residues 1–16), BR136 (244-257), Anti4R (275-291), BR135 (323-335), TauC4 (354–369) and BR134 (428-441) are underlined. (**B**) Representative immuno-EM images with antibodies BR133, BR136, Anti4R, BR135, TauC4, and BR134 of heparin-induced 2N4R and 2N3R tau filaments without (-) and with pronase (+) treatment. Scale bar, 100 nm. (**C**) Table summarising the results from B, and comparison with the immuno-EM results of AD and PiD. Tick marks indicate antibody decoration of filaments; crosses indicate absence of decoration. The four boxes where the human diseases differ from the in vitro heparin-induced filaments are highlighted in blue.

DOI: https://doi.org/10.7554/eLife.43584.014

The following figure supplement is available for figure 6:

**Figure supplement 1.** Western blots.
DOI: https://doi.org/10.7554/eLife.43584.015



**Figure 7.** Comparison of known tau filament structures. (**A**) β-strands and loop regions in the filaments are shown in different colours below the primary sequence of the microtubule-binding repeats (R1–R4). (**B**) Schematic representation of the different tau folds: the paired helical filament (PHF) and straight filament (SF) from Alzheimer's disease (AD); the narrow Pick filament (NPF) and wide Pick filament (WPF) from Pick's disease (PiD); the heparin-induced 2N4R snake (4 R-s), twister (4 R-t) and jagged (4 R-j); and the 2N3R heparin-induced filaments (3R). (**C**) Comparison of the structures of heparin-induced filaments of 2N4R and 2N3R tau with those of tau protofilaments from AD and PiD.

DOI: https://doi.org/10.7554/eLife.43584.016

in twister filaments could be due to their cross-β interfaces containing more polar groups. The interior packing of such groups requires the formation of hydrogen bonds. This may impose additional constraints on the mutual orientations of the opposite β-sheets, resulting in a larger packing angle between β-sheets, and thereby a larger helical twist.

The common, kinked hairpin fold among the three 2N4R structures may explain why 2N4R tau filaments can transition from one type into another. The observation that hose filaments, whose structure we could not determine, transition into snake and jagged filaments, suggests that they may adopt a similar, kinked hairpin conformation. Apparently, the energetic cost of the mismatch in β-sheet-forming hydrogen bonds is not large enough to preclude transitions from one filament type into another. These transitions are relatively rare, with only 15 observed transitions in 717 images. Nevertheless, the possibility of transitions occurring may complicate the interpretation of amyloid seeding experiments, which often assume replication of the seed conformation (*Jarrett and Lansbury, 1993*).

Heparin-induced filaments of 2N3R tau are less polymorphic than their 2N4R counterparts; 98% of all 3R filaments adopt a conformation where the ordered core comprises the third repeats of two parallel tau molecules. The presence of three β-strand breaking proline residues in the N-terminal region of R1 ([244]QTAPVPMPDL[253]) may explain why 2N3R tau does not form kinked hairpin-like structures, similar to the 2N4R tau filament types.

The structures of heparin-induced tau filaments explain a range of observations from biochemical and biophysical studies of filaments assembled in vitro that were inconsistent with the structures of tau filaments from AD and PiD. Limited proteolysis experiments (*Pérez et al., 2001*; *Santa-María et al., 2006*; *von Bergen et al., 2000*; *von Bergen et al., 2006*) indicated that both R2 and R3 become ordered in heparin-induced 4R tau filaments, whereas only R3 becomes ordered in 3R filaments. This agrees well with the ordered cores in the heparin-induced structures, but is inconsistent with the AD and PiD structures, which extend at least until F378. It is also incompatible with a widely used model system for in vitro tau aggregation, which uses the 4R-containing K18 fragment, comprising residues 244–372, and its 3R counterpart K19 (*Gustke et al., 1994*). Whereas K18 and K19 were at least six residues too short to adopt the same structures as those of tau filaments from AD and PiD, the ordered cores of the heparin-induced tau structures lie within the K18 and K19 sequences.

Assembly studies in vitro have shown that β-sheet formation in the first six residues of R2 ([275]VQIINK[280]) and/or the first six residues of R3 ([306]VQIVYK[311]) is important for filament formation of 4R tau (*Li and Lee, 2006*; *von Bergen et al., 2000*; *von Bergen et al., 2001*). In 3R tau, which lacks R2, only the hexapeptide at the beginning of R3 is present. Deletion of either hexapeptide motif reduces tau assembly, but only [306]VQIVYK[311] is necessary for filament formation (*Ganguly et al., 2015*; *Li and Lee, 2006*). It is present in the cores of all tau filaments from human brain whose high-resolution structures have been determined (*Falcon et al., 2018b*; *Fitzpatrick et al., 2017*). However, it packs with different residues in the different structures. In heparin-induced 3R tau filaments, [306]VQIVYK[311] packs with itself, reminiscent of the homotypic interactions seen in microcrystals (*Sawaya et al., 2007*). By contrast, in heparin-induced 4R tau filaments, this motif does not run throughout as a contiguous β-strand, but terminates with the conserved kink between Y310 and K311. Hexapeptide [275]VQIINK[280] of R2 is located in the fuzzy coat of AD filaments, but it is found in the cores of heparin-induced 4R tau filaments, where it forms a contiguous β-strand. It has been reported that VQIVYK inhibitors can block heparin-induced assembly of 3R, but not 4R, tau. Conversely, VQIINK inhibitors blocked heparin-induced assembly of 4R tau (*Seidler et al., 2018*). Our structures of heparin-induced 3R and 4R tau filaments are consistent with these findings.

Heparin-induced K19 tau filaments seed the assembly of both K19 and K18 tau monomers, whereas K18 filaments only seed assembly of K18 monomers (*Dinkel et al., 2011*; *Siddiqua et al., 2012*). Our structures, using 2N3R and 2N4R tau, provide an explanation for these observations. The 3R structure comprises mainly residues from R3, which is present in both 3R and 4R tau. If one were to incorporate a 4R tau molecule into the 3R dimer structure, only K274 would be replaced by S305. As K274 points outwards, at the edge of the ordered core, the energetic cost compared to incorporating a 3R tau molecule would be small. The reverse is not true. The 4R structure comprises the whole of R2. Therefore, incorporation of a 3R tau molecule in the 4R filament would implicate positioning R1 onto R2. Because half of the residues in R1 and R2 are different, including the three

additional prolines mentioned above, this would come at a higher energetic cost. By contrast, a recent study (*Weismiller et al., 2018*) has reported that sonicated filaments assembled from 2N4R human tau can seed assembly of monomeric 0N3R tau. It remains to be seen what the structures of those filaments are.

Nuclear magnetic resonance (NMR) studies of full-length tau and K18/K19 tau fragments indicated that R2 and R3 become ordered in heparin-induced 4R tau filaments (*Mukrasch et al., 2005*; *Sibille et al., 2006*), whereas only R3 becomes ordered in heparin-induced 3R tau filaments (*Andronesi et al., 2008*; *Daebel et al., 2012*; *Xiang et al., 2017*). A combination of hydrogen/deuterium exchange NMR, X-ray fibre diffraction and solid-state NMR showed that R3 peptides assemble into amyloids with two parallel R3 molecules, and with similar secondary structure elements as in our 2N3R structure (*Stöhr et al., 2017*). Moreover, in solid-state NMR experiments of heparin-induced filaments of K19, residues 321–324 exhibited two sets of resonances (*Daebel et al., 2012*), which is in agreement with our 2N3R structure. These residues engage in close interdigitating packing between β-strands of the opposing molecules, which includes the side-chains of S320, C322, S324 and the backbone of G326. Phosphates on S320 and S324 are incompatible with this structure, explaining why phosphorylation of these residues inhibits the heparin-induced assembly of 3R tau (*Schneider et al., 1999*).

Solid-state NMR also identified an intermolecular disulphide-bond between C322 residues in K19 filaments that were formed under oxidising conditions (*Daebel et al., 2012*). Under reducing conditions, filaments were still observed to form, albeit more slowly. Keeping tau in an oxidizing environment inhibits 4R filament formation, presumably through the formation of intermolecular disulphide bonds between C291 and C322. However, 3R tau readily formed filaments in an oxidizing environment, and filament formation was impaired under reducing conditions or when C322 was replaced with alanine (*Barghorn and Mandelkow, 2002*; *Schweers et al., 1995*; *Wille et al., 1992*). Whereas the AD and PiD structures did not explain the role of disulphide bond formation in tau aggregation, C322 residues in different molecules of the 3R dimer are within disulphide bonding distance of each other. Although we prepared our filaments under reducing conditions, it is likely that formation of a disulphide bond between those cysteine residues will facilitate the formation of similar filaments under oxidising conditions.

Tau is a soluble protein, and its bulk assembly requires polyanionic co-factors, such as heparin (*Goedert et al., 1996*; *Pérez et al., 1996*). Although our maps do not resolve heparin molecules with enough detail to build an atomic model, they do provide hints about the possible roles of heparin in filament formation. All four maps show fuzzy densities adjacent to lysines that point outwards from the filament cores. The positive charges of the lysines on many identical rungs of the cross-β helix need to be neutralised in order to form a stable filament. We hypothesize that this is a function of heparin; it may also cross-link tau molecules through their repeats (*Ramachandran and Udgaonkar, 2011*). Heparin is a polymer of a variably sulphated, repeating disaccharide unit. Whereas sulphation contributes negative charges, the disaccharide parts of the molecule can participate in both polar and non-polar interactions. This may explain the presence of a second type of fuzzy density, next to hydrophobic patches on the outside of the ordered cores of heparin-induced tau filaments (pink arrows in *Figures 2B*, *3B*, *4B* and *5B*). A model of charge compensation and possible transient incorporation of heparin into tau filaments is consistent with observations from nuclear magnetic resonance (NMR) (*Sibille et al., 2006*; *von Bergen et al., 2006*).

Perhaps the most important question is what the heparin-induced tau structures can teach us about filament formation in neurodegenerative diseases. Although the overall fold of heparin-induced filaments is different from the Alzheimer and Pick folds, there are also similarities between them. As in diseases, heparin-induced tau filaments are made of identical rungs of tau molecules that form cross-β structures by parallel stacking of identical β-strands along the helical axis. Perpendicular to the helical axis, the β-strands are interspersed with short loop regions, and all tau filament structures observed thus far share a common pattern of β-strand formation. The different folds mainly arise from differences in the loop regions, which result in packing otherwise similar β-strands against each other in different cross-β arrangements. It could still be that residues in R2 of filaments from tauopathies with 4R-only inclusions, like PSP, which are yet to be solved, may turn out to adopt a conformation that is similar to that of the heparin-induced 4R tau filaments. Mutations in this region, i.e. P301L, P301S and P301T, cause hereditary frontotemporal dementia with parkinsonism linked to chromosome 17 (FTDP-17T) (*Bugiani et al., 1999*; *Hutton et al., 1998*; *Lladó et al., 2007*;

*Poorkaj et al., 1998*). Mutations P301L and P301S were found to accelerate heparin-induced tau aggregation (*Barghorn et al., 2000*; *Goedert et al., 1999*). As proline residues interrupt hydrogen bond interactions across the rungs, the position of P301 in the partially disordered hammerhead arc may cause its disordered structure. This could explain why replacing this proline with leucine or serine facilitates filament formation by stabilising the local structure.

A striking difference between the heparin-induced tau structures and those from diseases is in the charge distribution on the outward-facing residues. Whereas both Alzheimer and Pick folds contain stretches of outward-facing residues with alternating positive and negative charges, the heparin-induced structures are more positively charged. This difference mainly arises from the presence of more negative residues in R4, which is part of the Alzheimer and Pick folds, but which is disordered in the heparin-induced structures. We hypothesize that by neutralising positive charges on the filaments, the negative charges in heparin allow the formation of in vitro structures that would not be stable in the brain.

However, the Alzheimer and Pick folds still contain outward-facing residues for which a positive charge is not compensated by a close-by, negatively charged residue. Therefore, cofactors or post-translational modifications of tau may be required for filament formation in the brain. For example, acetylation of lysines could reduce positive charges on the filaments. Alternatively, polyanionic molecules in neurons could perform a similar role as heparin. In multiple cases of AD, filament structures showed similar fuzzy densities in front of lysines, reminiscent of those attributed to heparin in the structures described here (*Falcon et al., 2018a*). We previously hypothesized that neutralising, negative charges could also be provided by residues in the fuzzy coat (*Fitzpatrick et al., 2017*). In particular, we highlighted [7]EFE[9], which is part of the structural epitope of Alz50 and MC-1 antibodies (*Jicha et al., 1997*). It could be that these residues also play a role in stabilizing heparin-induced tau filaments (*Bibow et al., 2011*).

Our results demonstrate that a single protein, in this case tau, can adopt many different amyloid conformations. Whereas similar residues form β-strands among the different structures, the turns and loops between the β-strands, as well as the side-chain interactions between opposing β-sheets, are very versatile. This leads to highly variable cross-β packings and helical parameters. Moreover, the observation that in vitro assembly may yield filaments that are different from those found in human neurodegenerative diseases calls for caution when interpreting structures from in vitro systems. The structural versatility we observe for tau filaments may also occur for other assemblies. For example, negative stain EM imaging revealed differences between filaments extracted from tissues with systemic amyloidosis and those of the same proteins assembled in vitro (*Annamalai et al., 2017*). It therefore remains to be shown if the in vitro assembled amyloid structures of, for example, amyloid-β (*Gremer et al., 2017*) and α-synuclein (*Guerrero-Ferreira et al., 2018*; *Li et al., 2018a*; *Li et al., 2018b*) are the same as those in human brain. Therefore, atomic structures of filaments extracted from human tissues are eagerly awaited for more proteins and diseases. Discovering what drives the formation of different types of amyloid filaments in different diseases will be crucial for our understanding of, and possibly, our ability to intervene in disease.

# Materials and methods

## Key resources table

| Reagent type (species)or resource | Designation | Source or reference | Identifiers | Additional information |
|---|---|---|---|---|
| Recombinant DNA | Plasmid: pRK172-2N4R | PMID: 2124967; 8849730; 9407097 | NCBI Reference Sequence: NM_005910.5 | Plasmid can be provided upon reasonable request. |
| Recombinant DNA | Plasmid: pRK172-2N3R | PMID: 2124967 | NCBI Reference Sequence: NM_001203252.1 | Plasmid can be provided upon reasonable request. |
| Strain, strainback ground (E. coli) | BL21 (DE3) | Agilent Technologies | 200131 | |
| Chemical compound, drug | Heparin | Sigma-Aldrich | H4784 | |

*Continued on next page*

*Continued*

| Reagent type (species)or resource | Designation | Source or reference | Identifiers | Additional information |
|---|---|---|---|---|
| Chemical compound, drug | Chymostatin | Sigma-Aldrich | C7268 | Protease inhibitor |
| Antibody | BR133 (Anti-N- terminus of tau proteins, Rabbit polyclonal) | In house PMID: 28678775; 30158706 | | WB dilution: 1:4000 EM dilution: 1:50 |
| Antibody | BR134 (Anti-C- terminus of tau proteins, Rabbit polyclonal) | In house PMID: 28678775; 30158706 | | WB dilution: 1:4000 EM dilution: 1:50 |
| Antibody | BR136 (Anti-R1 of tau proteins, Rabbit polyclonal) | In house PMID: 30158706; 30276465 | | WB dilution: 1:4000 EM dilution: 1:50 |
| Antibody | Anti-4R (Anti-R2 of 2N4R tau protein, Rabbit polyclonal) | Cosmo Bio PMID: 28678775; 30158706; 30276465 | CACTIP4RTP01 | WB dilution: 1:2000 EM dilution: 1:50 |
| Antibody | BR135 (Anti-R3 of tau proteins, Rabbit polyclonal) | In house PMID: 28678775; 30158706; 30276465 | | WB dilution: 1:4000 EM dilution: 1:50 |
| Antibody | TauC4 (Anti-R4 of tau proteins, Rabbit polyclonal) | Masato Hasegawa PMID: 28678775; 30158706; 30276465 | | WB dilution: 1:2000 EM dilution: 1:50 |
| Software, algorithm | RELION | PMID: 30412051 | RRID:SCR_016274 | |
| Software, algorithm | COOT | PMID: 20383002 | RRID:SCR_014222 | |
| Software, algorithm | REFMAC | PMID: 15299926 | RRID:SCR_014225 | |
| Software, algorithm | PHENIX | PMID: 20124702 | RRID:SCR_014224 | |

WB: Western Blot; EM: Electron microscopy.

## Tau expression and purification

Tau was expressed and purified as described (*Bugiani et al., 1999*; *Hasegawa et al., 1998*), with some modifications. The cDNAs coding for human 2N4R and 2N3R tau were cloned into pRK172, which was transformed into *Escherichia coli* BL21 (DE3). Cells were cultured in 2xTY medium supplemented with 5 mM MgCl$_2$ and 100 mg/l ampicillin at 37°C until an OD600 of 0.8, when expression was induced by addition of 0.4 mM isopropyl-1-thio-β-D-galactopyranoside. After 3 hr, cells were collected by centrifugation, resuspended in buffer A (50 mM MES pH6.5, 50 mM NaCl, 10 mM EDTA, 5 mM MgCl$_2$, 5 mM TCEP, 0.1 mM AEBSF, 0.03 mM Chymostatin supplemented with cOmplete EDTA-free Protease Inhibitor Cocktail (Roche)) and lysed by ultrasonication (Sonics VCX-750 Vibra Cell Ultra Sonic Processor, 3 min of working time, 3 s on, 6 s off, at 40% amplitude). After incubation with 40 µg/ml DNAse (Sigma) and 10 µg/ml RNAse (Sigma) for 5 min, the lysates were centrifuged at 15,000 × g for 30 min at 4°C. Supernatants were loaded onto a Hitrap CaptoS column (GE Healthcare) and eluted with a 50–500 mM NaCl gradient. Peak fractions were analysed by Tris-Glycine SDS-PAGE (4–20%) and stained with Coomassie brilliant blue R250 (Fisher Chemical). The purified fractions were pooled and precipitated with 38% ammonium sulphate. The pellets were resuspended in buffer B (PBS plus 5 mM TCEP, 0.1 mM AEBSF, 0.015 mM Chymostatin supplemented with cOmplete EDTA-free Protease Inhibitor Cocktail) and centrifuged at 100,000 × g at 4°C for 1 hr. The supernatants were loaded onto a pre-equilibrated HiLoad 16/60 Superdex 200 column (GE Healthcare) with buffer B and eluted at a flow rate of 1 ml/min. Fractions were pooled and concentrated to 3.0 mg/ml. Aliquots of purified protein were snap-frozen and stored at −20°C.

## Heparin-induced filament assembly of tau

Tau proteins (3.0 mg/ml) were incubated with heparin (400 µg/ml, 6–30 kDa, Sigma) in 30 mM MOPS, pH 7.2, 1 mM AEBSF; and 4 mM TCEP at 37°C for 3 days, as described (*Goedert et al., 1996*). The molar ratio of tau:heparin was approximately 4:1.

## Electron cryo-microscopy

Before making cryo-grids, the heparin-induced assembly reactions were centrifuged at 100,000 g for 30 min at 4°C. The resulting pellets were resuspended in 20 mM Tris, pH 7.4, 100 mM NaCl. Pronase-treated tau filaments (3 µl, at 2.0 mg/ml) were applied to glow-discharged holey carbon grids (Quantifoil Au R1.2/1.3, 300 mesh), blotted with filter paper and plunge-frozen in liquid ethane using an FEI Vitrobot Mark IV. For 2N4R filaments, imaging was performed on an FEI Tecnai G2 Polara microscope operating at 300 kV using a Falcon III detector prototype in integrating mode. A total of 717 movies of 30 frames was recorded during 1.0 s exposures, at a pixel size of 1.38 Å on the specimen, and a total dose of approximately 48 e/Å². Defocus values ranged from −1.7 to −2.8 µm. For 2N3R filaments, imaging was performed on a Gatan K2-Summit detector in counting mode, using an FEI Titan Krios at 300 kV. A GIF-quantum energy filter (Gatan) was used with a slit width of 20 eV to remove inelastically scattered electrons. A total of 2051 movies of 44 frames was recorded during 11 s exposures, at a pixel size of 1.04 Å on the specimen, and a total dose of 50 electrons per Å². Defocus values ranged from −0.8 to −2.2 µm. Further details are presented in *Table 1*.

## Helical reconstruction for the 2N4R filaments

Movie frames were gain-corrected, aligned, dose weighted and then summed into a single micrograph using MOTIONCOR2 (*Zheng et al., 2017*). Aligned, non-dose-weighted micrographs were used to estimate the contrast transfer function (CTF) using CTFFIND4.1 (*Rohou and Grigorieff, 2015*). All subsequent image-processing steps were performed using helical reconstruction methods in RELION 3.0 (*He and Scheres, 2017*; *Scheres, 2012*; *Zivanov et al., 2018*). Each of the four types of filaments was selected manually in the micrographs, and the resulting data sets were processed independently.

For snake filaments, 303,754 segments were extracted with an inter-box distance of 14 A° and a box size of 600 pixels. Initial reference-free 2D classification was performed with images that were down-scaled to 128 pixels to speed up calculations. Segments contributing to suboptimal 2D class averages were discarded. Assuming a helical rise of 4.7 A°, a helical twist of −1.4° was estimated from the crossover distance of filaments in the micrographs. Using these parameters, an initial 3D reference was reconstructed from the 2D class averages de novo. We then re-extracted the selected segments without down-scaling them, and with a smaller box size of 256 pixels. Using these segments and the de novo initial model low-pass filtered to 20 A°, we performed two rounds of 3D classification with six classes, each time selecting the segments contributing to the best 3D class for subsequent 3D auto-refinement with optimisation of helical twist and rise. A final 3D auto-refinement of 52,441 selected segments converged onto a helical twist of −1.26°. The helical rise was kept fixed at 4.70 Å. The corresponding reconstruction was sharpened with a B-factor of −41.26 Å² (*Table 1*), using the standard post-processing procedure in RELION. For model building, helical symmetry was imposed on the post-processed map using the RELION helix toolbox (*He and Scheres, 2017*). The overall resolution of the final map was estimated as 3.3 Å from Fourier shell correlations at 0.143 between the two independently refined half-maps, using phase-randomization to correct for convolution effects of a generous, soft-edged solvent mask (*Chen et al., 2013*). Local resolution estimates were obtained using the same phase-randomization procedure, but with a soft spherical mask that was moved over the entire map.

Twister and jagged filaments were processed in a similar manner. Twister segments were initially extracted in 800 pixel boxes that were downsized to 128 pixels for reference-free 2D class averaging. 4R-jagged segments were extracted in 600 pixel boxes and down-scaled to 256 pixels. The initial estimates for the helical twist, as estimated from crossover distances in the micrographs, were −3.5° for twister and −2.1° for jagged filaments. These values were again used for de novo calculation of initial 3D models from the 2D class averages. For both data sets, final segments were extracted in boxes of 256 pixels without down-scaling, and 3D classification was used to select the best segments. CTF refinement and Bayesian polishing in RELION-3.0 were used in an attempt to further increase the signal-to-noise ratio in the segments (*Zivanov et al., 2018*). Detailed parameters for both data sets are reported in *Table 1*.

For hose filaments, 124,458 segments were extracted using a box size of 1200 pixels, and down-scaled to 384 pixels to speed up 2D classification. Similar to the other 4R filament types, de novo initial model generation was attempted from the reference-free 2D class averages. However, possibly

due to the large degree of bending and an apparent lack of twist in many filaments, all 3D reconstruction attempts failed.

## Helical reconstruction for the 2N3R filaments

Processing of the 2N3R dataset was similar to the 2N4R dataset, but CTF parameters were estimated using Gctf (*Zhang, 2016*) instead of CTFFIND4.1, and 788,359 segments were selected using automated picking procedures for helices in RELION-3.0 (*He and Scheres, 2017*). Segments were initially extracted with a box size of 800 pixels and down-scaled to 256 pixels for reference-free 2D class averaging. Two types of 2D class averages were observed, corresponding to narrow and wide filaments. We only proceeded with the narrow filaments, as they comprised 98% of the segments. An initial helical twist of −1.1° was estimated from the crossover distance of filaments in micrographs, and used for de novo 3D initial model calculation from the 2D class averages. Parameters for the final reconstruction are given in *Table 1*.

## Model building and refinement

Atomic models were built de novo in the maps with imposed helical symmetry using COOT (*Emsley et al., 2010*). Model building was started from a distinctive feature of the 3R filament: a cross-β packing with a very short distance between the β-sheets that can only be achieved for residues with small or no side chains at the interface. In the tau sequence, there is only one segment that could form a β-strand with four sufficiently small, inwards facing residues, [320]SKCGSLG[326] from R3, making this sequence assignment unambiguous. Extension of the sequence towards the N- and C-terminal regions, by manually adding amino acids in COOT, confirmed this assignment with other distinctive residues, like the large aromatic side chains of Y310, matching their clear densities. The assignment of lysine side-chains on the filament surface brought their ε-amino groups close to the observed external diffuse densities, presumably corresponding to the sulphate groups of heparin. This observation then also allowed ready identification of lysine residues in the 4R tau filament structures. The clear densities of the di-lysine [317]KVTSK[321] motifs, combined with good densities for other bulky side chains like Y310, provided the starting point for complete sequence assignment.

The four structures are devoid of strong handedness, and, at the reported resolutions, it is not possible to determine their absolute hand based on densities for carbonyl groups of the main chain. For the snake filaments, we assumed the same handedness for the [290]KCGSKD[295] motif as we observed for the homologous [353]KIGSLD[358] motif in the tau filament structures from AD (*Fitzpatrick et al., 2017*). This corresponded to a negative twist angle, similar to that observed for AD filaments. The direction of twist of the other three structures was then kept the same as for the snake filaments. Initial manual model building was followed by targeted real-space refinement in COOT. The model was then translated to give a stack of three consecutive monomers to preserve nearest-neighbour interactions for the middle chain in subsequent refinements using a combination of rigid-body fitting in COOT and Fourier-space refinement in REFMAC (*Murshudov et al., 1997*). Because most residues adopted a β-strand conformation, hydrogen-bond restraints were imposed to preserve a parallel, in-register hydrogen-bonding pattern in earlier stages of Fourier-space refinements. Local symmetry restraints were imposed to keep all β-strand rungs identical. Side-chain clashes were detected using MOLPROBITY (*Chen et al., 2010*), and corrected by iterative cycles of real-space refinement in COOT and Fourier-space refinement in REFMAC and PHENIX (*Adams et al., 2010*). For each refined structure, separate model refinements were performed against a single half-map, and the resulting model was compared to the other half-map to confirm the absence of overfitting. The final models were stable in refinements without additional restraints. Statistics for the final models are shown in *Table 1*.

## Immunolabelling

Western blotting and immuno-EM were carried out as described (*Falcon et al., 2018b*; *Fitzpatrick et al., 2017*; *Goedert et al., 1992*). For immuno-EM, pronase treatment was performed by incubating filaments with 0.4 mg/ml pronase (Sigma) for 1 hr at 21°C. Blocking used PBS and 0.5% BSA. Primary and secondary antibodies were used at 1:50 and 1:20, respectively.

## Acknowledgements

We thank Christos Savva, Giuseppe Cannone, Joanna Brown and Shaoxia Chen at the MRC-LMB EM facility for help with cryo-EM; Jake Grimmet and Toby Darling for help with high-performance computing, and Masato Hasegawa for antibody TauC4. We also thank Yuriy Chaban and Diamond for access and support of the Cryo-EM facilities at the UK electron Bio-imaging Centre (eBIC), (proposal EM17434-17), funded by the Wellcome Trust, MRC and BBSRC, for data acquisition of the 2N3R data set. MG is an Honorary Professor in the Department of Clinical Neurosciences of the University of Cambridge. This work was supported by the UK Medical Research Council (MC_U105184291 to MG and MC_UP_A025_1013 to SHWS) and the European Union (Joint Programme-Neurodegeneration Research REfrAME) to MG and BF EU/EFPIA/Innovative Medicines Initiative [2] Joint Undertaking (IMPRIND grant n° 116060) to MG. WZ is supported by a Foundation that prefers to remain anonymous.

## Additional information

### Competing interests

Sjors HW Scheres: Reviewing editor, *eLife*. Michel Goedert: Reviewing editor, *eLife*. The other authors declare that no competing interests exist.

### Funding

| Funder | Grant reference number | Author |
|---|---|---|
| European Union | Joint Programme - Neurodegeneration Research REfrAME | Benjamin Falcon Michel Goedert |
| Medical Research Council | MC_U105184291 | Michel Goedert |
| European Union | IMPRIND-116060 | Michel Goedert |
| Medical Research Council | MC_UP_A025_1013 | Sjors HW Scheres |

The funders had no role in study design, data collection and interpretation, or the decision to submit the work for publication.

### Author contributions

Wenjuan Zhang, Conceptualization, Formal analysis, Investigation, Visualization, Writing—original draft, Writing—review and editing; Benjamin Falcon, Conceptualization, Formal analysis, Investigation, Writing—original draft, Writing—review and editing; Alexey G Murzin, Conceptualization, Formal analysis, Validation, Investigation, Visualization, Writing—original draft, Writing—review and editing; Juan Fan, Investigation, Methodology; R Anthony Crowther, Conceptualization, Formal analysis, Investigation, Writing—review and editing; Michel Goedert, Conceptualization, Supervision, Investigation, Writing—original draft, Writing—review and editing; Sjors HW Scheres, Conceptualization, Software, Formal analysis, Supervision, Funding acquisition, Validation, Investigation, Visualization, Methodology, Writing—original draft, Project administration, Writing—review and editing

### Author ORCIDs

Wenjuan Zhang http://orcid.org/0000-0002-3011-9956
Sjors HW Scheres http://orcid.org/0000-0002-0462-6540

### Decision letter and Author response

Decision letter https://doi.org/10.7554/eLife.43584.039
Author response https://doi.org/10.7554/eLife.43584.040

# Additional files

## Supplementary files

• Transparent reporting form

DOI: https://doi.org/10.7554/eLife.43584.017

## Data availability

EM maps have been submitted to EMDB, under codes 4563, 4564, 4565 and 4566. Atomic models have been submitted to PDB under codes 6QJH, 6QJM, 6QJP and 6QJQ. Raw EM images have been submitted to EMPIAR under codes 10242 and 10243.

The following datasets were generated:

| Author(s) | Year | Dataset title | Dataset URL | Database and Identifier |
|---|---|---|---|---|
| Wenjuan Zhang, Benjamin Falcon, Alexey G Murzin, Juan Fan, R Anthony Crowther, Michel Goedert | 2019 | Cryo-EM reconstruction of heparin-induced 2N4R tau snake filaments | https://www.ebi.ac.uk/pdbe/entry/emdb/EMD-4563 | Electron Microscopy Data Bank, EMD-4563 |
| Wenjuan Zhang, Benjamin Falcon, Alexey G Murzin, Juan Fan, R Anthony Crowther, Michel Goedert, Sjors HW Scheres | 2019 | Cryo-EM reconstruction of heparin-induced 2N3R tau filaments | https://www.ebi.ac.uk/pdbe/emdb/empiar/entry/10242 | EMPIAR, 10242 |
| Wenjuan Zhang, Benjamin Falcon, Alexey G Murzin, Juan Fan, R Anthony Crowther, Michel Goedert, Sjors HW Scheres | 2019 | Cryo-EM reconstruction of heparin-induced 2N4R tau twister filaments | https://www.ebi.ac.uk/pdbe/entry/emdb/EMD-4564 | Electron Microscopy Data Bank, EMD-4564 |
| Wenjuan Zhang, Benjamin Falcon, Alexey G Murzin, Juan Fan, R Anthony Crowther, Michel Goedert, Sjors HW Scheres | 2019 | Cryo-EM reconstruction of heparin-induced 2N4R tau jagged filaments | https://www.ebi.ac.uk/pdbe/entry/emdb/EMD-4565 | Electron Microscopy Data Bank, EMD-4565 |
| Wenjuan Zhang, Benjamin Falcon, Alexey G Murzin, Juan Fan, R Anthony Crowther, Michel Goedert, Sjors HW Scheres | 2019 | Cryo-EM reconstruction of heparin-induced 2N3R tau filaments | https://www.ebi.ac.uk/pdbe/entry/emdb/EMD-4566 | Electron Microscopy Data Bank, EMD-4566 |
| Wenjuan Zhang, Benjamin Falcon, Alexey G Murzin, Juan Fan, R Anthony Crowther, Michel Goedert, Sjors HW Scheres | 2019 | Cryo-EM structure of heparin-induced 2N4R tau snake filaments | https://www.rcsb.org/structure/6QJH | RCSB Protein Data Bank, 6QJH |
| Wenjuan Zhang, Benjamin Falcon, Alexey G Murzin, Juan Fan, R Anthony Crowther, Michel Goedert, Sjors HW Scheres | 2019 | Cryo-EM structure of heparin-induced 2N4R tau twister filaments | https://www.rcsb.org/structure/6QJM | RCSB Protein Data Bank, 6QJM |
| Wenjuan Zhang | 2019 | Cryo-EM structure of heparin-induced 2N4R tau jagged filaments | https://www.rcsb.org/structure/6QJP | RCSB Protein Data Bank, 6QJP |
| Wenjuan Zhang, | 2019 | Cryo-EM structure of heparin- | https://www.rcsb.org/ | RCSB Protein Data |

| Benjamin Falcon, Alexey G Murzin, Juan Fan, R Anthony Crowther, Michel Goedert, Sjors HW Scheres | | induced 2N3R tau filaments | structure/6QJQ | Bank, 6QJQ |
| Wenjuan Zhang, Benjamin Falcon, Alexey G Murzin, Juan Fan, R Anthony Crowther, Michel Goedert, Sjors HW Scheres | 2019 | Cryo-EM reconstruction of heparin-induced 2N4R tau filaments | https://www.ebi.ac.uk/pdbe/emdb/empiar/entry/10243 | EMPIAR, 10243 |

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
