## [Decision Letter]

Thank you for submitting your article "Heparin-induced tau filaments are polymorphic and differ from those in Alzheimer's and Pick's diseases" for consideration by *eLife*. Your article has been reviewed by three peer reviewers, including Nikolaus Grigorieff as the Reviewing Editor and Reviewer #1, and the evaluation has been overseen by Cynthia Wolberger as the Senior Editor. The following individuals involved in review of your submission have agreed to reveal their identity: Henning Stahlberg (Reviewer #2); Marcus Fändrich (Reviewer #3).

The reviewers have discussed the reviews with one another and the Reviewing Editor has drafted this decision to help you prepare a revised submission.

Summary:

There is agreement among the reviewers that the new data and results presented by the authors are important for Alzheimer's and Pick's disease research. As pointed out by the authors, the difference in structures observed in ex vivo and in vitro filaments must be taken into account in Alzheimer's/Pick's disease models. However, the observation that ex vivo and in vitro amyloid filaments can differ in morphology is perhaps not that surprising since this has been observed also for other amyloid filaments, e.g. by Annamalai et al. (2017), which the authors could cite as an example.

The present work also describes other, more interesting and generally relevant results that warrant greater emphasis and discussion in the manuscript. These include the high-resolution reconstructions of several filament morphologies formed by the same peptide, which enable for the first time a detailed comparison of how the peptide folds and side chain interactions differ between these morphologies. What can we learn from these structures about polymorphism, as well as common elements in amyloid filaments (see detailed comments below)? Furthermore, the observation that a specific morphology/fold can switch to another morphology/fold within a single filament has fundamental implications on experiments that use seeds to induce specific morphologies, an important paradigm for many studies in the amyloid field. Arguably, this result is therefore also more relevant to Alzheimer's/Pick's research than the described differences between ex vivo and in vitro filaments. The authors should offer some ideas for how this can happen and explain what this means for seeding experiments in general. An expanded discussion of these two aspects (polymorphism, fold switching) will give the present work broader relevance and justify its publication in *eLife*.

---

## [Author Response]

Summary:There is agreement among the reviewers that the new data and results presented by the authors are important for Alzheimer's and Pick's disease research. As pointed out by the authors, the difference in structures observed in ex vivo and in vitro filaments must be taken into account in Alzheimer's/Pick's disease models. However, the observation that ex vivo and in vitro amyloid filaments can differ in morphology is perhaps not that surprising since this has been observed also for other amyloid filaments, e.g. by Annamalai et al. (2017), which the authors could cite as an example.

We have added this example to the last paragraph of the Discussion.

The present work also describes other, more interesting and generally relevant results that warrant greater emphasis and discussion in the manuscript. These include the high-resolution reconstructions of several filament morphologies formed by the same peptide, which enable for the first time a detailed comparison of how the peptide folds and side chain interactions differ between these morphologies. What can we learn from these structures about polymorphism, as well as common elements in amyloid filaments (see detailed comments below)?

We have made several modifications to place more emphasis on the general implication of structural versatility of amyloids.

The last sentence of the Abstract now reads:

"Our results illustrate the structural versatility of amyloid filaments, and raise questions about the relevance of in vitro assembly."

The first paragraph of the Discussion now reads:

"Cryo-EM structures of three of these conformations reveal a common, kinked hairpin fold, with differences in kink, helical twist and offset distance of the ordered core from the helical axis. […] This may impose additional constraints on the mutual orientations of the opposite β-sheets, resulting in a larger packing angle between β-sheets, and thereby a larger helical twist."

And the last paragraph of the Discussion now reads:

"Our results demonstrate that a single protein, in this case tau, can adopt many different amyloid conformations. […] Discovering what drives the formation of different types of amyloid filaments in different diseases will be crucial for our understanding of, and possibly, our ability to intervene in disease."

Furthermore, the observation that a specific morphology/fold can switch to another morphology/fold within a single filament has fundamental implications on experiments that use seeds to induce specific morphologies, an important paradigm for many studies in the amyloid field. Arguably, this result is therefore also more relevant to Alzheimer's/Pick's research than the described differences between ex vivo and in vitro filaments. The authors should offer some ideas for how this can happen and explain what this means for seeding experiments in general. An expanded discussion of these two aspects (polymorphism, fold switching) will give the present work broader relevance and justify its publication in eLife.

We have added the following, now second, paragraph to the Discussion:

"The common, kinked hairpin fold among the three 2N4R structures may explain why 2N4R tau filaments can transition from one type into another. […] Nevertheless, the possibility of transitions occurring may complicate the interpretation of amyloid seeding experiments, which often assume replication of the seed conformation (Frost el al., 2009; Guo et al., 2016)."